# The Addition of Poly(Vinyl Alcohol) Fibers to Apatitic Calcium Phosphate Cement Can Improve Its Toughness

**DOI:** 10.3390/ma12091531

**Published:** 2019-05-10

**Authors:** Jun Luo, Julien Faivre, Håkan Engqvist, Cecilia Persson

**Affiliations:** Division of Applied Materials Science, Department of Engineering Sciences, Uppsala University, Box 534, 751 21 Uppsala, Sweden; luojuncd@scu.edu.cn (J.L.); julien.faivre@etu.emse.fr (J.F.); hakan.engqvist@angstrom.uu.se (H.E.)

**Keywords:** fiber reinforcement, apatite cement, poly(vinyl alcohol), composite, compressive strength, work of fracture, diametral tensile strength, toughness

## Abstract

Calcium phosphate cements, and in particular hydroxyapatite cements, have been widely investigated for use as bone void fillers due to their chemical similarity to bone and related osteoconductivity. However, they are brittle, which limits their use to non-load-bearing applications. The aim of the current study was to improve the toughness of hydroxyapatite cements through fiber reinforcement. The effect of the addition of hydrophilic, poly(vinyl-alcohol) (PVA) fibers to hydroxyapatite cement was evaluated in terms of mechanical properties, including compressive strength, diametral tensile strength and toughness (work of fracture), as well as setting time, phase composition and cement morphology. The fiber reinforcement enhanced the fracture resistance of the hydroxyapatite cement, but also simultaneously reduced the compressive strength and setting time of the cements. However, cement with 5 wt % of fibers (of the powder component) could be considered a good compromise, with a compressive strength of 46.5 ± 4.6 MPa (compared to 62.3 ± 12.8 MPa of that without fibers), i.e., still much greater than that of human trabecular bone (0.1–14 MPa). A significantly higher diametral tensile strength (9.2 ± 0.4 MPa) was found for this cement compared to that without fibers (7.4 ± 1.5 MPa). The work of fracture increased four times to 9.1 ± 1.5 kJ/m^2^ in comparison to the pristine apatite. In summary, the hydroxyapatite cements could be reinforced by suitable amounts of PVA fibers, which resulted in enhancing the material’s structural integrity and ductility, and increased the material’s resistance to cracking.

## 1. Introduction

Calcium phosphate cements (CPCs) are extensively investigated materials for bone replacement applications. They are prepared by mixing a solid and a liquid phase to form a paste that hardens over time. The solid phase is composed of one or more calcium phosphate powders and the liquid phase generally consists of distilled water that may or may not contain additives [1,2]. There are two main types of CPCs: acidic cement resulting in monetite or brushite (dicalcium phosphate dihydrate) and basic cement resulting in apatite (hydroxyapatite (HA) or calcium-deficient hydroxyapatite). Both types are moldable, osteoconductive and—to different extents—biodegradable [3], having chemical similarities with the calcium phosphates found in human bone, which motivated their use as a bone void fillers [2]. Apatite cements typically have a very low degradation rate and have traditionally been considered stronger than acidic cements [4]. Nevertheless, they remain brittle materials with a low fracture toughness and are only indicated for non-load-bearing applications [5,6]. 

To extend the application of apatite cements to load-bearing or at least load-sharing applications, many researchers have tried to improve their mechanical properties. One of the most successful strategies is to mitigate their brittle behavior by using biocompatible fiber reinforcement to form composite materials [6]. The improvement can be attributed to the complex interaction between all of the composite constituents, in which the macroscopic mechanical behavior results from the mechanical properties of both the fiber and cement matrix and their mechanical interaction [7]. Different types of fibers, both polymeric (e.g., chitosan [8]) and ceramic (e.g., calcium silicate fibers [9]), degradable as well as non-degradable, have been extensively used to improve the toughness and ductility of apatite cements [9,10]. However, cements reinforced with biodegradable polymer fibers usually exhibit low elastic moduli and strength, and have not been used in load-bearing applications [7]. For example, biodegradable PLGA 910 fibers have been evaluated in several studies [11,12,13], but the fibers lost their strength too fast due to the dissolution of the fibers in an aqueous environment, leaving the cement very porous and brittle (7 MPa in flexural strength after approx. 1.5 months [11], whereas the complete resorption may take 3 to 36 months [14]). 

Non-degradable fibers, such as aramid, have been used in CPC matrices to achieve stable cement composites with small changes in the mechanical properties [10]. Therefore, combining polymer fibers of a lower degradability with CPCs may be a valuable route to their reinforcement. Poly(vinyl alcohol) (PVA) is generally considered a biocompatible material, with mechanical properties and a degradability depending on factors such as processing, molecular weight, and crystallinity [15,16]. It also has a high affinity to water [17]. This hydrophilicity could be advantageous when used in water-based matrices such as CPCs, giving a better integration with the matrix. In fact, PVA fibers have been found to increase the tensile strength and ductility without excessively decreasing the compression strength of a calcium aluminate cement [18]. However, PVA for fiber-reinforcement of CPCs has not yet been studied.

The aim of this study was therefore to evaluate the effect of PVA fibers on apatite cement in terms of its mechanical properties, namely its compressive strength (CS), diametral tensile strength (DTS), and work of fracture. Setting time measurements, XRD, and SEM analysis were also performed to determine the optimum amount of PVA fibers. 

## 2. Materials and Methods

### 2.1. Materials

α-TCP powder was acquired from RMS Foundation, Switzerland. The particle size distribution (in volume average) was d10 = 2.7 ± 0.1 µm, d50 = 9.4 ± 1.0 µm and d90 = 87 ± 27 µm [19]. PVA fibers were purchased from STW (Type F PVA 401–100, Schenkenzell, Germany) and reported by the manufacturer to have a tensile modulus of 40 GPa and a tensile strength of 1.83 GPa. The fibers were 0.9 mm in length and 13 µm in diameter. Sodium phosphate dibasic (Na_2_HPO_4_) was bought from Sigma Aldrich (St. Louis, MO, USA). 

### 2.2. Cement Preparation

The PVA fibers were mixed into the α-TCP powder by using a shaker-mixer (Turbula T2F, WAB, Muttenz, Switzerland) at 62 rpm for 2.5 hours. Different proportions of fibers were incorporated and tested; 0%, 2.5%, 5%, and 7.5% of the total weight of the final powder mixture. Whilst 10 wt% of fiber reinforcement was also tested, this paste was not workable and set too fast. The control group without PVA fibers will henceforth be referred to as group CPC, and the groups with 2.5, 5, and 7.5 wt % of PVA fiber will be referred to as group CPC-2.5 PVA, CPC-5.0 PVA, and CPC-7.5 PVA respectively. The liquid phase was comprised of 2.5 wt % Na_2_HPO_4_ solution, and a liquid-to-powder (and fiber) ratio of 0.35 mL/g was used. The powder and liquid phases were mixed in a cap vibrator (Ivoclar Vivadent, AGFL, Schaan, Principality of Liechtenstein) for 1 min. The paste was transferred into different rubber molds with a spatula within 5 min from the start of mixing. To perform the diametral tensile tests, samples in rubber molds of 8 mm in diameter and 3.5 mm in height were prepared. For the compressive test, samples in 6 mm diameter and 13 mm height molds were prepared. The cements were allowed to set for 1 h at room temperature (22 ± 1 °C) before being stored in 0.1 M PBS solution (P4417-100TAB, Sigma-Aldrich, St. Louis, MO, USA) for 7 days at 37 °C. Samples were polished to acquire flat and parallel surfaces with 1200 grit silicon carbide paper before mechanical tests. After polishing, samples for the CS test were 12 mm in height (6 mm in diameter). In the same way, samples for DTS had a height of 3 mm (8 mm in diameter).

### 2.3. Setting Time Test

For the setting time test, a Gillmore Apparatus was used in conformity with ASTM C266-03 [20].

### 2.4. Mechanical Testing

The mechanical properties, including CS and DTS, were measured using a universal testing machine (AGS-X, Shimadzu, Kyoto, Japan) at a cross-head speed of 1mm/min and a pre-load of 1 N. At least six specimens were tested in each group. The DTS was calculated by DTS = $\frac{2P}{\pi Dt}$, where *P* is the applied load, *t* the thickness of the disc, and *D* the diameter [21,22]. From the diametral tensile test, the work of fracture (the energy absorbed by the specimens before the fracture) was calculated from the area under the load-displacement curve, up to 1 mm displacement, and normalized by the specimen’s cross-sectional area [23]. The specimen data was corrected for machine compliance by subtracting the machine compliance (measured from a compression test without a specimen) from each specimen’s apparent compliance [24]. The real stiffness of the specimen is calculated by taking the inverse of the compliance. 

### 2.5. Microstructural Analysis

The samples from the CS testing were used to analyze the microstructure. Prior to analysis, the samples were dried in a vacuum for 24 h. The fracture surfaces of dry samples were analyzed with scanning electron microscopy (SEM, LEO 1550, Zeiss, Oberkochen, Germany) under an accelerating voltage of 3.00 kV with an SE2 detector. The samples were previously sputtered with a thin Au/Pd coating for 30 s.

### 2.6. Compositional Analysis

After vacuum drying for 24 hours, dried samples were ground to a fine powder and analyzed by X-ray diffraction (XRD, D8 Advance, Bruker AXS GmbH, Karlsruhe, Germany) using Cu-K_α_ irradiation in a theta-theta setup and a 2 θ range from 5 to 60° in steps of 0.02° with 0.25 s/step and a rotation speed of 80 rpm. A quantitative analysis by Rietveld refinement with BGMN software (www.bgmn.de) [25,26] in combination with Profex (http://profex.doebelin.org [27]) was performed. The reported result was the mean of three measurements. The crystalline models used in the refinement were α-TCP from PDF# 04-010-4348 [28] and HA from PDF# 01-074-0565 [29]. 

### 2.7. Statistical Analysis

Analysis of variance was performed with IBM® SPSS® Statistics v. 22 (IBM Corp., Chicago, IL, USA). A critical level of α = 0.05 was used. Since homogeneity of variance could not be confirmed for all groups (Levene’s test), Welch’s robust test of equality of means and Tamhane’s post-hoc test were used.

## 3. Results

### 3.1. Mechanical Properties

Typical mechanical responses of the cements are shown in Figure 1. After reaching the highest value, the stress of the unreinforced cements drastically decreased for both compressive stress-strain curves and diametral tensile stress-strain curves, indicative of catastrophic failure, while that of the cements with increasing amounts of fibers showed an increasingly moderate decrease in the two curves. The mechanical properties of the cements, as derived from the CS tests, are summarized in Figure 2. As shown in Figure 2a, CPC samples demonstrated the highest CS (62.3 ± 12.8 MPa). The CS decreased with increasing fiber content, with the samples in group CPC-7.5 PVA showing the lowest CS (35.3 ± 4.4 MPa). There was no significant difference between the CS of group CPC and group CPC-2.5 PVA. As shown in Figure 2b, the Young’s modulus of group CPC (7.3 ± 2.1 GPa) was similar to that of group CPC-2.5 PVA (7.7 ± 0.7 GPa). However, similarly to the CS, the Young’s modulus of group CPC varied more than the fiber-containing cements, as shown by the larger standard deviation. Groups CPC-5.0 PVA and CPC-7.5 PVA showed similar Young’s modulus, but significantly lower than that of group CPC-2.5 PVA (Figure 2b). No significant differences among the four groups could be seen for their failure strain, measured as the strain at the first drop in load (Figure 2c). Figure 3 shows the mechanical properties of the cements from the DTS tests. The lowest DTS was found for group CPC (7.4 ± 1.5 MPa), while group CPC-5.0 PVA exhibited the highest value of DTS at 9.2 ± 0.4 MPa (Figure 3a). The DTS of group CPC-7.5 PVA (8.0 ± 0.5MPa) was significantly lower than groups CPC-2.5 PVA and CPC-5.0 PVA. As shown in Figure 3b, with the addition of PVA fibers, the work of fracture, as determined from the DTS tests, also increased notably from 2.5 ± 0.7 kJ/m^2^ to 6.8 ± 2.2 kJ/m^2^, 9.1 ± 1.5 kJ/m^2^ and 9.5 ± 0.9 kJ/m^2^ for group CPC, CPC-2.5 PVA, CPC-5.0 PVA, and CPC-7.5 PVA, respectively. There was no significant difference between the work of fracture of group CPC-5.0 PVA and CPC-7.5 PVA. Moreover, after the matrix had cracked, in the presence of PVA fibers, the specimens in group CPC-2.5 PVA, CPC-5.0 PVA, and CPC-7.0 PVA had similar fracture behavior (i.e., they were still intact due to fibers bridging the cracks) and did not completely break into pieces as seen with group CPC (Figure 4). 

### 3.2. Setting Time

The samples in group CPC demonstrated the longest initial (24.0 ± 2.0) and final (84.0 ± 7.0) setting times (Table 1). The setting time decreased with an increasing amount of fibers. The samples of group CPC-7.5 PVA showed the shortest initial (8.5 ± 1.0) and final (53 ± 9.0) setting time. 

### 3.3. Microstructure

The SEM images of cross-sectional fracture surfaces are shown in Figure 5. For the samples with PVA fibers, fibers as well as the voids due to the pull-out of fibers, appeared to be evenly distributed inside the apatite matrix. As shown in Figure 5e,f, at a higher magnification, the morphologies of plate-like apatite crystals in group CPC and group CPC-5.0 PVA are similar, and apatite crystals were in close contact with the fibers. 

### 3.4. Composition

As shown in Table 2 and Figure 6, the inorganic composition of each group showed similar amounts of apatite in the final cements (about 97 wt %).

## 4. Discussion

The clinical indication of CPCs is restricted to non-load bearing applications, in particular because of their intrinsic brittleness [6]. In this study, the incorporation of PVA fibers into apatite cement in order to improve its toughness was evaluated in terms of the resulting CS, Young’s modulus, failure strain, DTS, toughness (work of fracture), setting time, microstructure, and inorganic composition. 

According to the compressive stress-strain curves and diametral tensile stress-strain curves (Figure 1), the unreinforced cements exhibited the characteristic catastrophic failure, attributed to intrinsic brittleness, while the cements with increasing amounts of fibers could resist higher strains, albeit at a lower load. The CS of the apatite cement decreased with increasing amount of fibers (Figure 2a), which might be attributed to fiber aggregation and the flexible PVA fibers. However, the lowest value of CS (35.3 ± 4.4 MPa), in group CPC-7.5 PVA, was still significantly higher than that of human trabecular bone, which has been reported to vary between 0.1–14 MPa [30,31]. Only the CPC-5.0 PVA and CPC-7.5 PVA groups, with the higher amount of PVA fibers, led to a significant reduction of Young’s modulus. As expected there were no significant differences in failure strains between the groups from the CS tests (Figure 2c), since these were taken at the first load drop, reflecting mainly the failure stress of the matrix. However, the addition of PVA fibers delayed a complete failure, and more so with increasing amount of PVA fibers (Figure 1a). The improvement of the DTS with PVA fibers (Figure 3a) demonstrated that the load was transferred through the apatite matrix to the PVA fibers. Whilst the DTS increased with increasing amount of PVA fibers, any further increase was limited by the fiber aggregation resulting from the addition of excessive amounts of PVA [7]. The mechanical property enhancement was even more evident in the work of fracture results (Figure 3b). Indeed, the work of fracture increased by a factor of four and pictures taken after the mechanical tests (Figure 4) demonstrated that the addition of PVA fibers transformed the brittle apatite cement into a more ductile composite [32,33]. Besides the decrease in compressive strength, the addition of PVA fibers resulted in apatite cements which better resist fracture and prevent the propagation of cracks in the samples. With 2.5 wt % PVA fiber reinforcement, the CS and DTS remained unchanged, whereas the work of fracture was multiplied by three and was statistically significantly different from the pristine apatite cement. It was therefore demonstrated that the addition of hydrophilic PVA fibers can prevent complete failure of the composite even after multiple cracks by fiber bridging and fiber pull-out [7]. 

The compressive strength and Young’s modulus of PVA-reinforced cements were higher than some other fiber-reinforced calcium phosphate cements (e.g., collagen reinforced apatite cement [34] and similar to some others (e.g., carbon nano tube/reinforced apatite cements [35]) [6,7]. The DTS and the work of fracture calculated from the DTS were not reported in most studies on fiber reinforcement, since bending tests (three or four point bending) were commonly reported instead [7]. However, because of the bar-shaped, bigger samples for bending tests, the noise in the results from these tests might be important, especially when the strains are very small, like for CPCs [36]. The combination of hydrophilic PVA fibers and water-based apatite matrix showed good mechanical strength, and the high tensile strength, good biocompatibility and stability of PVA fibers may be beneficial for long-term applications, such as prophylactic femeroplasty.

The setting time of the cements decreased with increasing fiber content, from 24 min initial setting time for the pristine cement to 8.5 minutes for that containing 7.5 wt % fibers. This was most likely due to the PVA absorbing water [37], leading to an actual lower liquid–to-powder ratio in the setting reaction. This did not, however, seem to affect the setting of the apatite cement, as a similar composition of the apatite was demonstrated in the XRD phase composition analysis, for all groups.

The mechanical test results are further supported by the microstructural analysis, which elucidates disparities in results and shows lines of improvement. Relatively long fibers were sticking out of the surface, indicating a superior fracture strength of the fiber relative to the shear stresses created during the test. This could be counteracted by increasing the diameter or the length of the fibers, or by improving the interfacial adhesion between fibers and matrix, e.g., through enhancement of the wettability and reactivity of fibers [6]. The microstructure of the apatite crystals in all samples was found to be similar to a previous study [38], with no differences between groups, again confirming that the PVA fibers did not interfere with the formation of apatite. 

There are some limitations to the current study. One limitation is the lack of injectability of the cements – they could not be injected through a syringe in a simple test. Hence, the cements may either not be used in minimally invasive surgery or need to be further optimized (e.g., shorter fibers). However, it can be noted that for applications such as femoroplasty, a low injectability may not necessarily be a concern, as large drill holes in strategic patterns may be used [39]. Furthermore, there is a lack of knowledge on the degradation rate of these cements. This type of PVA fiber is likely to have a very low degradation rate, but the study can be considered a proof-of-concept for further development of similar composites with an optimized degradation rate. Finally, while extensively used in the research field of CPCs, the DTS test may underestimate their strength [40]. Therefore, a direct tensile test or a bending test may be recommended for future studies. In the next steps, standard fracture tests would also be performed on the samples to further confirm the resistance to cracking due to the addition of PVA fibers. Other key properties of the composites would also be investigated, including fatigue properties, the adhesion strength between cement and fibers, effect of the fiber geometry (length, diameter), effect of fiber surface properties on the composites, the degradation behavior, and biocompatibility.

## 5. Conclusions

In this study, the toughness of apatite cement was enhanced with PVA fibers. The PVA fibers did not appear to interfere with the formation of apatite. The optimum amount of PVA fibers was found to be 5 wt % for the formulations investigated herein, which gave the cement a compressive strength of 46.5 ± 4.6 MPa, Young’s modulus of 5.2 ± 0.9 GPa, diametral tensile strength of 9.2 ± 0.4 MPa, and a work of fracture of 9.1 ± 1.5 kJ/m^2^ (in DTS), i.e., rendering the cement more ductile and avoiding the catastrophic failure mode typically seen in non-reinforced CPCs. The use of PVA fibers shows the potential for load-bearing or load-sharing applications and could be a promising alternative for bone replacement applications, as it increases the material’s resistance to cracking. 

## Figures and Tables

**Figure 1 materials-12-01531-f001:**
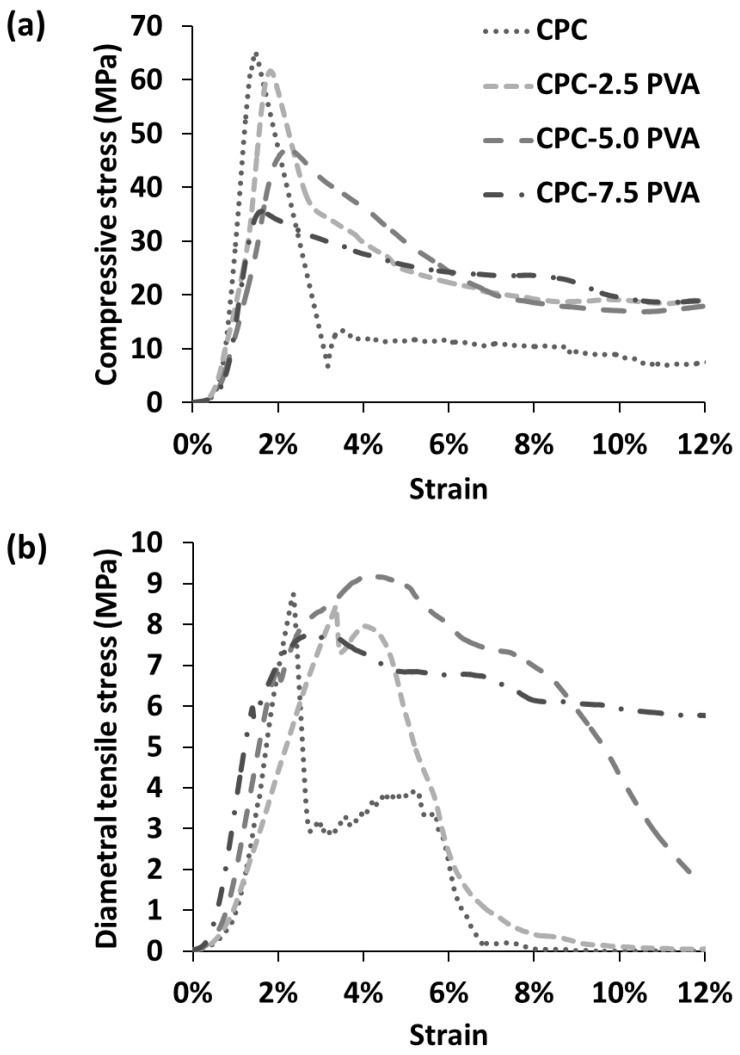
Typical mechanical response of the cements. (**a**) Compressive stress-strain curves; (**b**) diametral tensile stress-strain curves. One curve out of 6–9 samples per group is shown for clarity.

**Figure 2 materials-12-01531-f002:**
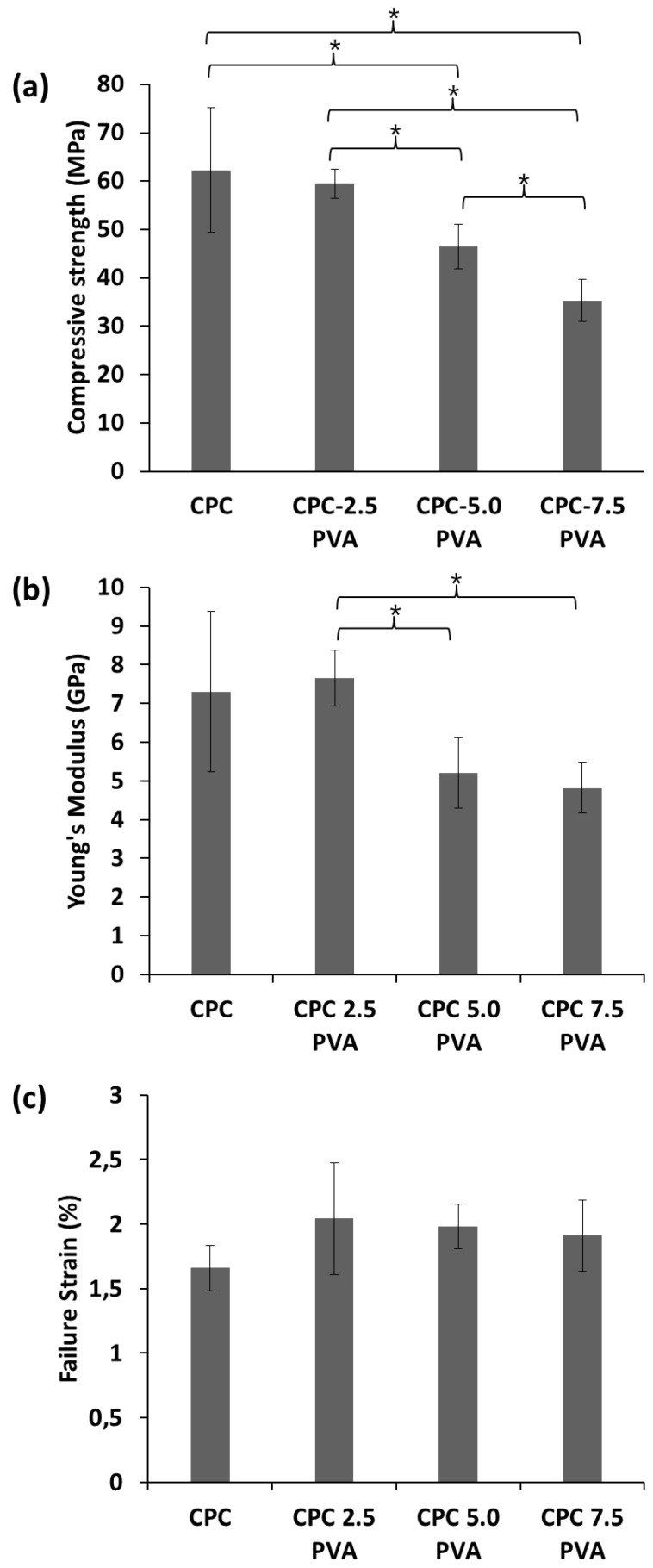
Mechanical properties of the cements from compressive strength (CS) tests. (**a**) Compressive strength; (**b**) Young’s modulus; (**c**) failure strain (taken at first load drop). * Statistically significant difference (ANOVA, Scheffe’s post-hoc test, *p* < 0.05).

**Figure 3 materials-12-01531-f003:**
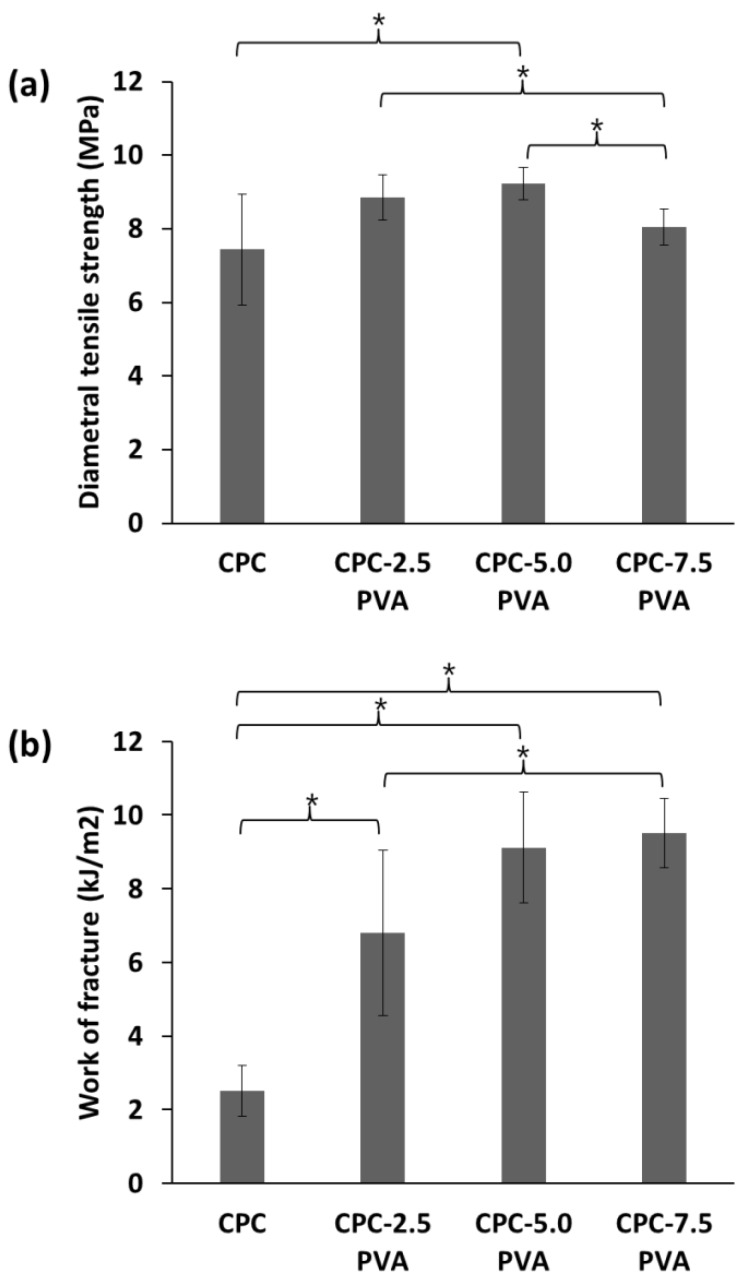
Mechanical properties of the cements from diametral tensile strength (DTS) tests. (**a**) Diametral tensile strength; (**b**) work of fracture. * Statistically significant difference (ANOVA, Scheffe’s post-hoc test, *p* < 0.05).

**Figure 4 materials-12-01531-f004:**
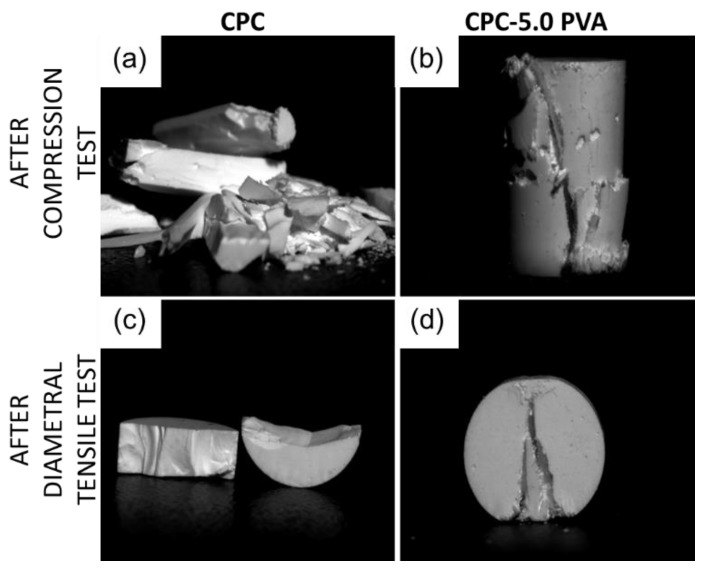
Pictures of the samples after compression of (**a**) group CPC and (**b**) group CPC-5.0 PVA; and after the diametral tensile test of (**c**) group CPC and (**d**) group CPC-5.0 PVA.

**Figure 5 materials-12-01531-f005:**
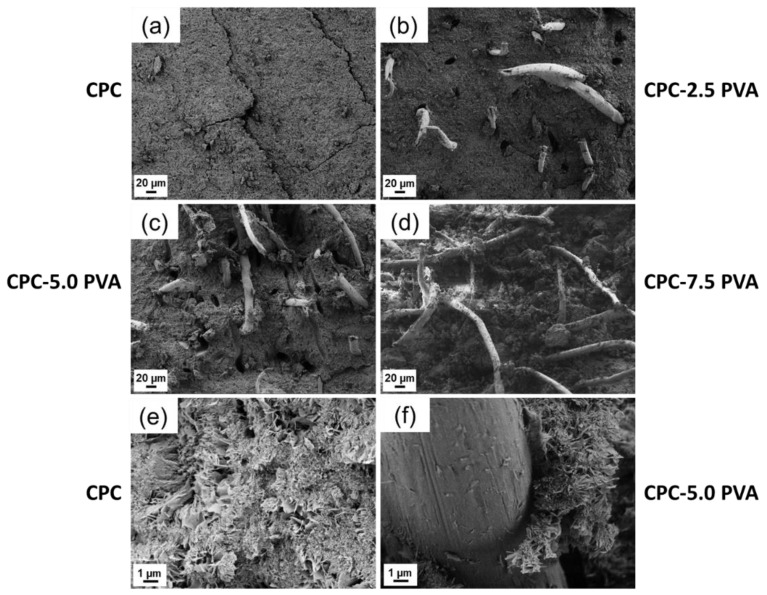
SEM micrographs of fractured surfaces of the cements. (**a**) Group CPC (0 wt % PVA); (**b**) group CPC-2.5 PVA (2.5 wt % PVA); (**c**) group CPC-5.0 PVA (5 wt % PVA); (**d**) group CPC-7.5 PVA (7.5 wt % PVA). (**e**) Group CPC and (**f**) group CPC-5.0 PVA at different magnifications.

**Figure 6 materials-12-01531-f006:**
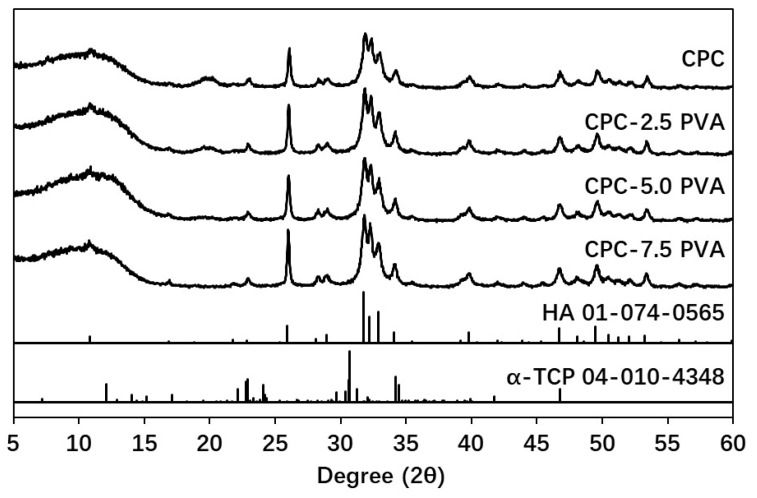
Representative XRD patterns of samples (one of three measurements is shown per group) and the reference Powder Diffraction Files (PDFs).

**Table 1 materials-12-01531-t001:** Setting time of the cements (average ± standard deviation).

Group	Initial (min)	Final (min)
CPC	24.0 ± 2.0	84 ± 7.0
CPC-2.5 PVA	13.5 ± 2.0	77.0 ± 9.0
CPC-5.0 PVA	11.5 ± 1.0	64.0 ± 6.0
CPC-7.5 PVA	8.5 ± 1.0	53.0 ± 9.0

**Table 2 materials-12-01531-t002:** Inorganic phase composition of the cements (average ± standard deviation).

Group	α-TCP (wt %)	Apatite (wt %)
CPC	3.2 (± 0.1)	96.8 (± 0.1)
CPC-2.5 PVA	2.6 (± 0.3)	97.4 (± 0.3)
CPC-5.0 PVA	3.1 (± 0.7)	96.9 (± 0.7)
CPC-7.5 PVA	2.7 (± 0.3)	97.3 (± 0.3)

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
