# Peer review of "The Addition of Poly(Vinyl Alcohol) Fibers to Apatitic Calcium Phosphate Cement Can Improve Its Toughness"

_materials, 2019, doi:10.3390/ma12091531_

Round 1
Reviewer 1 Report
The manuscript submitted me for review entitled " The addition of poly(vinyl alcohol) fibres to apatitic calcium phosphate cement can improve its toughness" is well organized and carried out. The article contains original and significant results based on well conducted experiments and measurements. The results are well discussed and the conclusion is adequate to them. I have not found any substantive errors. Therefore I recommend the manuscript for publication in Materials in the present form.
Author Response
Dear Editor,
We want to thank you and the reviewers for taking our manuscript into consideration and providing valuable comments. We have carefully read each comment and responded to them accordingly.
Reviewer 2 Report
This is now ready for publication.
Author Response

(The authors gave the same response as above.)

Reviewer 3 Report
The manuscript reports the study of the properties of PVA fibres reinforced experimental calcium phosphate cements. The manuscript is thoroughly written and is certainly of interest to the community. I only have couple of minor comments to the authors.
1) I would recommend to include the examples of the actual XRD patterns.
2) In the 2.4 Mechanical Testing section, the last sentence indicates that some data corrections were applied for machine compliance. Would the authors kindly consider explain it more specifically or give some reference.
3) In the section 3.3 Microstructure, the figure number referred to in the text should be corrected to 5e and 5f.
Author Response
Dear Editor,
We want to thank you and the reviewers for taking our manuscript into consideration and providing valuable comments. We have carefully read each comment and responded to them accordingly.
Below we comment on each point raised by the reviewer and summarize the changes made in the manuscript. Our responses are marked in red. Changes in the manuscript are highlighted throughout the text.
Point 1: I would recommend to include the examples of the actual XRD patterns.
Response 1: We have added examples of the actual XRD patterns as Figure 6.
Point 2: In the 2.4 Mechanical Testing section, the last sentence indicates that some data corrections were applied for machine compliance. Would the authors kindly consider explain it more specifically or give some reference.
Response 2: We have provided further explanation for the machine compliance correction and added reference 25:
“The specimen data was corrected for machine compliance by subtracting the machine compliance (measured from a compression test without a specimen) from each specimen’s apparent compliance [25]. The real stiffness of the specimen is calculated by taking the inverse of the compliance. ”
Point 3: In the section 3.3 Microstructure, the figure number referred to in the text should be corrected to 5e and 5f.
Response 3: We have corrected the figure number in the text.